# Welfare Problems in Cattle, Pigs, and Sheep that Persist Even Though Scientific Research Clearly Shows How to Prevent Them

**DOI:** 10.3390/ani8070124

**Published:** 2018-07-20

**Authors:** Temple Grandin

**Affiliations:** Department of Animal Science, Colorado State University, Fort Collins, CO 80523, USA; Cheryl.miller@colostate.edu

**Keywords:** welfare, cattle, swine, lameness, at risk cattle, bovine respiratory disease

## Abstract

**Simple Summary:**

Great strides have been made to improve animal welfare. Unfortunately, there are certain problems that continue to persist. The causes of these problems range from a lack of financial accountability for losses, failure to measure them, or repeating old mistakes. Some examples of persistent problems are bruises, failure to vaccinate cattle, and high percentages of lame livestock. Both good management practices and providing the right financial incentives will improve welfare.

**Abstract:**

Poor production and handling practices continue to persist that are both detrimental to animal welfare and financially burdensome. These practices continue to persist for three reasons: (1) a segmented marketing chain where a producer is not held financially accountable for losses; (2) failure to measure and assess chronic painful problems such as lame livestock; and (3) repeating old mistakes, such as housing fattening cattle for long periods of time on bare concrete. Two examples of the first type of losses are bruises caused by poor handling and sick cattle at feedlots caused by failure to vaccinate and precondition weaned calves at the farm of origin. In some segmented marketing systems, there is no economic incentive to vaccinate. When the animals get sick, the responsibility gets passed to the next person. Buyers of meat products can reduce these “passed on” losses by source verification. The first step to reducing problems, such as lame livestock, is to measure the percentage of lame animals and work with the producers to reduce them. Also, transportation payments should be changed and contracts should be based on the condition of the animals at delivery.

## 1. Introduction 

Great strides have been made in animal welfare research. Both scientific studies and practical experience clearly show that best practices in both stockmanship and housing will produce more productive animals [1,2]. Unfortunately, some producers do not follow the recommendations made by researchers. Why do certain bad practices still persist today when they are detrimental to animal welfare and cost money? The poor practices continue even though producers know they are wrong because there is not a sufficient financial incentive to stop doing them. For example, bruises on beef cattle cost the cattle industry millions of dollars worldwide. Poor handling and transportation practices are a major cause of bruises. Improving cattle handling and transportation practices can reduce the percentage of carcasses with bruises [3,4]. Another problem that has resurfaced in recent years is raising fed cattle on concrete slats which results in swollen joints [5]. This causes problems with excessive electric goad use at the abattoir. Under EU legislation, severely lame animals cannot be transported. In the United States, there are no regulations to prohibit transport of severely lame animals.

The practice of shipping weaned calves from pasture to feedlots with no vaccinations still occurs today. This is a bad practice that will increase both morbidity and mortality [6,7]. Research clearly shows that vaccinating and weaning beef calves 45–60 days before they leave the ranch of origin will reduce sickness and death losses [8,9]. Vaccinating calves on the day they are shipped is worthless. The vaccine will not have time to create immunity before the calves are exposed to pathogens at the feedlot. It has been suggested that the feedlot veterinarians should make vaccinations a requirement before entry. The United States has very little legislation regarding this matter and it would be difficult to implement anyway. If one feedlot required preconditioning and vaccinations, the producer may send the calves to a feedlot that does not have this requirement.

Lameness is another serious issue in dairy cows, pigs, and poultry. Lame animals are less productive [10]. There are some dairies that do an excellent job of preventing their cows from becoming lame and others with high percentages of lame cows. Research clearly shows that there are huge differences in the percentage of lame cows between the best dairies and the worst dairies [11,12].

The purpose of this paper is to discuss why these problems continue to persist and address some ways to remedy these problems. There are four major causes of these persistent animal welfare problems. They are: (1) lack of financial accountability for losses; (2) poor management; (3) wrong financial incentives; and (4) “bad becoming normal”. The latter occurs when conditions deteriorate slowly and people do not perceive the problem.

## 2. Repeating an Old Mistake—Lame Cattle Housed on Concrete

High percentages of lame cattle is a good example of bad becoming normal. In the United States, problems with lameness in fed steers and heifers have recently increased. This is partly due to expansion of the ethanol industry, which has provided a financial motivation to feed cattle in the Midwest close to where the corn (maize) is grown. Wet distillers grain from the ethanol industry is fed to cattle. Since this feedstuff is expensive to ship, cattle that had previously been fed in dry western dirt feedlots are now remaining in the midwestern United States. Due to high rainfall, some cattle feeders are now fattening cattle on bare concrete floors without bedding. Dirt feedlots in the Midwest become extremely muddy due to high rainfall. The good cattle feeders provide bedding in the concrete pens, but some producers raise cattle on bare concrete. This can result in both swollen knee joints [5] and lameness. There have been reports from people working in abattoirs that the worst lame fed cattle arrive during the night shift. Managers are less likely to see them at this time. Cattle that are reluctant to move are more likely to be abused with electric goads. In the 1970s, the author observed three slatted feedlots with bare concrete slats. In all three feedlots, swollen joints and lameness started occurring after the animals had gained 225 kg on the slats. A rubber surface placed on the slats slows down the development of swollen joints [5], but it does not eliminate leg injuries. Unfortunately, some US cattle feeders are now repeating the same mistakes they made in the 1970s. At that time, huge feedlots with concrete slatted floors were built in the United States. During the 1980s, they were phased out due to lameness, other health issues, and high costs. The author has been in the beef industry for over 40 years. New people coming into the industry are reinventing old bad mistakes because they do not know about them.

The author has also observed recent problems with lame cattle in other countries that house fattening cattle on bare concrete. High rainfall or scarce bedding materials are the reason for this poor practice. The practice persists because there is no economic incentive to either provide bedding or shorten the length of time that the cattle are kept on bare concrete.

## 3. No Financial Accountability for Bruised, Dead, or Sick Cattle

Failure to vaccinate weaned beef calves before they leave the ranch, bruises caused by rough handling, and overloading of trucks are often caused by a lack of accountability for losses. Cattle that are moved through auctions or are handled poorly have more bruises than cattle sold directly from the farm to abattoir [13,14,15]. In this segmented market, no one is held financially accountable for bruises. When producers have to pay for bruises, they can be greatly reduced [3].

In the US cattle industry, a rancher who fails to vaccinate his or her weaned calves is often not held financially accountable if unvaccinated calves get sick or die at the feedlot. The calves often pass through a segmented market chain which usually consists of a local auction, order buyer (dealer), and a final destination at a feedlot. The calves often cross state lines and there is no Federal legislation for individual identification. In some countries, truckers are paid based on how many kilograms of livestock they can load onto a truck [16]. This provides a financial incentive to overload the truck.

## 4. Methods to Improve Financial Accountability

The first step is for major retail and restaurant buyers to require source-verified livestock. This will make it possible for buyers to provide either financial rewards or fines for high levels of bruises or sick calves. Source verification back to the farm of origin is essential for the effective use of financial incentives. In Europe, legislated source verification procedures make it easier to trace animals back to the farm of origin. Many countries do not have legislated source verification. In the United States, progressive buyers have set up private programs to purchase only source-verified cattle. These animals will have been produced according to the buyer’s welfare and production guidelines [16]. In some states, special auctions are used to sell cattle to feedlots that have been verified as prevaccinated. Depending upon cattle-availability market prices, special sales of properly preconditioned cattle do not always result in higher prices for producers [17]. The special preconditioned auctions market the calves to cattle feedlots.

These voluntary private special auctions work well when cattle supplies are plentiful. They are less effective when cattle are scarce because cattle feeders need to fill up their yards. When this happens, greater numbers of “at-risk” cattle are sold at auctions. At-risk cattle is a euphemism for poorly managed calves which have not been prevaccinated. After arrival at the feedlot, the mass treating of at-risk cattle with antibiotics will lower death losses [18]. This improves the welfare of the calves but it is a poor practice due to increasing concerns about antibiotic resistance.

To improve accountability for losses, the author suggests the following recommendations:(1)High-welfare schemes, such as RSPCA Freedom Foods in the United Kingdom and Niman Ranch and American Humane Certified in the United States, require producers to perform best practices such as vaccinating calves before shipment from the ranch. Producers are encouraged to join these programs to get financial premiums. These programs are monitored by auditors who are independent from the livestock industry.(2)Retail and restaurant buyers of meat and dairy products should increase the purchase of livestock and poultry that can be source verified back to the farm of origin.(3)Abattoir management should provide both rewards and fines to reduce bruises and death losses. The author observed a great reduction in downed nonambulatory pigs arriving at an abattoir after initiating a $25 handling fee for each nonambulatory pig.(4)Change transportation payment and contracts based on the condition of the animals at delivery. Never use contracts based on the number of animals loaded at the farm of origin.

## 5. Problems Caused by Poor Management

When the author first started her career in the 1970s, she mistakenly thought that newer, more advanced equipment could be designed that would prevent poor handling. Equipment is only half the equation and good management is the other half. Too often people want the wonderful new technology that they think will solve all their problems. The author has a saying that “People want the thing more than the management”. New technology does not replace the need for good management. The purchase of a new technology, such as a fancy milking equipment, a new truck, or a computer, does not replace management. Purchase of a new technology is a one-time expense, whereas good management and stockmanship require continuous attention to many small details. Some of the details of procedure that require monitoring during handling are the number of animals moved in each group, electric goad use, and training employees to use behavioral principles of livestock handling.

## 6. Importance of Stockmanship

Research studies clearly show the benefits of good stockmanship. Fearful animals with large flight zones that avoid people are less productive [19,20,21]. During the last 10 years, the use of training programs to improve stockmanship has increased. Two research studies in the United States showed that cattle handling in large feedlots had improved [21,22]. A study done in California on ranches showed that ranchers who had Beef Quality Assurance training had better handling scores [23,24].

## 7. Ways to Improve Management

(1)Managers must care about animal welfare. Top managers who do not care will have problems in their operations. Audits of abattoirs showed that conditions improved when poor managers were removed [25].(2)Use outcome based numerical standards. Numerical scoring of handling practices greatly improved handling and stunning in slaughter plants when it was implemented by major meat buyers [25,26].(3)Do not understaff or overwork employees. Tired overworked employees will not care and are more likely to handle animals roughly.(4)Top managers must get out of the office and regularly observe conditions on farms and abattoirs.(5)Never pay people who handle livestock on a piecework basis. When a piecework program is used, the employee pay is based on the number of animals handled. Create incentives to reward employees for good practices. Reward employees handling livestock and poultry for low levels of death losses, injuries, and bruising.(6)Employees and stockpeople should be trained to have a positive attitude towards animals. Research studies clearly show that a positive attitude and liking animals improves productivity [1,27].

## 8. Bad Becomes Normal

A North American study published in 2012 reported that lameness in dairy cows became so severe that a quarter of all cows were classified as lame [28] and 33% were at risk of becoming lame [29]. This is just one example of bad becoming normal. The percentage of lame cows had slowly increased and it was not noticed by the dairy managers. Studies have also shown that a dairy manager will greatly underestimate the percentage of lame cows [30,31].

The first step in reducing lameness is to start measuring it. Research in Wisconsin has shown great progress in reducing the percentage of lame dairy cows [32]. This was accomplished by measuring lameness on a regular basis plus a comprehensive program of university extension personnel working with dairy managers. Studies have also shown that cows with swollen hocks were more likely to become lame. Good management practices, such as frequent attention to bedding or cubicles (freestalls), will reduce the percentage of cows with swollen hocks [33].

Another example of bad becoming normal is heat stress in large heavy feedlot cattle. Open-mouth breathing in cattle when they are at rest is a sign of severe heat stress [34,35]. Problems with heat stress may have increased because fed cattle are fed to heavier weights, reach market weight at younger ages, and are often black-hided cattle. Other issues may be the use of beta-agonists during hot weather [16,36]. The problem is that managers may get accustomed to looking at panting and think it is like a dog panting. Cattle that are panting are experiencing severe heat stress [34].

The bulldog is not a farm animal, but it is an extreme example of bad becoming normal. The breed standard is verbal. The American Kennel Club Official Standard States a bulldog should have a “massive short faced head, wide shoulders, and sturdy limbs”. When a verbal standard is followed, breeders do not know when to stop breeding animals with shorter and shorter noses. There is a picture from a 1938 New York Times article labelled “Bull dog’s dilemma”. It shows a functional dog. It is totally different from some of today’s extreme bulldogs.

## 9. Methods to Prevent Bad from Becoming Normal

Lameness—Use both lameness scoring systems and leg conformation assessments [37,38,39]. Continuous measurement will prevent the percentage of lame animals from gradually increasing.

Body Condition Scoring—This will prevent breeding livestock from becoming too thin [40]. For sows, a simple way to assess poor body condition is: if vertebrae bumps show along her back, she is too thin [16]. A sow that is showing vertebrae bumps would be fit for transport and she should be culled from the herd before she becomes unfit.

Injury Scoring and Bruise Scoring—Scoring systems are available for bruises on carcasses [41,42] and for other injuries, such as swollen hocks on dairy cows and confined beef cattle [33]. On sows, shoulder lesions, damage from fighting, foot lesions, and other abnormalities should be scored [43,44].

Livestock Hygiene Scoring—Dirty Animal Scorings—some good information is in Welfare Quality [45,46].

Scoring of Livestock Handling—Some of the variables that are scored are: electric goad use [47], slips and falls [45], and vocalization during handling and turning back or balking during handling [45,48,49]. The advantage of these scoring systems is that they make it possible to determine if practices are improving or becoming worse. Some systematic reviews of animal-based welfare indicators are in [50,51,52].

## 10. Abuse of Animals with Little Economic Value

Animals that have little economic value provide no economic incentive for treating them well. Some of the problem areas are old cull breeding stock and newborn bull dairy calves. In the United States, development of a large Holstein fed steer market has increased the value of bull dairy calves. In some parts of the United States, Holsteins are half of the fed cattle market. Dairies are also gradually changing their mindset that cattle are raised for both milk and beef. Developing methods to increase the economic value of an animal will often improve welfare.

## 11. Conclusions

There are welfare problems that still persist, such as unvaccinated cattle arriving at feedlots and high percentages of lame cattle. The first problem could be remedied by encouraging more producers to enroll in high-welfare programs that require best practices. High percentages of lame dairy cows is an example of bad becoming normal. This occurred because the percentage of lame dairy cows slowly increased without being noticed. The use of numerical scoring of lameness is recommended to help managers reduce lameness in both dairy and beef cattle.

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
