# Peer review of "Welfare Problems in Cattle, Pigs, and Sheep that Persist Even Though Scientific Research Clearly Shows How to Prevent Them"

_animals, 2018, doi:10.3390/ani8070124_

Round 1

Reviewer 1 Report

This invited review presents an accurate but depressing story of continuing abuses to the welfare of farm animals, especially beef cattle, despite clear understanding both among scientists and within the industry, of ways in which these abuses can be minimised.  Most of the problems described relate to practices current in the USA.  Some solutions are offered, e.g. financial accountability for unvaccinated beef calves or injured pigs but no evidence is presented to suggest that these are having a major impact.  In Europe and some states within the USA we have seen the development of ‘High Welfare’ schemes such as the RSPCA Freedom Food scheme and several schemes operated by independent retailers (supermarket chains).  The best of these schemes can achieve quality control (as distinct from mere quality assurance) through verification of source, independent structured monitoring of animal welfare and honest communication between producers, retailers and consumers.  I believe this review could be improved – and made more constructive - through inclusion of some discussion of progress with schemes, properly monitored by bodies independent of the production industry, especially in the USA.

Author Response

Reviewer 1 – Per reviewer 1’s suggestion:

Added an additional recommendation that producers should participate in high welfare programs such as RSPA Freedom Foods in the UK and Niman Ranch in Humane Certified in the U.S.

Reviewer 2 Report

This is a short article on a few welfare issues that still persist with author's opinion on how these problems could be addressed based on the many years of personal experience and seeing the changes of the industry.

Ln 19 Reword "they are somebody else's problem" to something like responsibility get's passed to the next person

Abstract Add a sentence at the end for fixing problem #3 outlined

Ln 28 Needs a "however" statement to transition from research is great but it's not being followed

Ln 41 Could the feedlot vets make vaccinations a requirement before entry? They have oversight and responsibility of the health status all the animals at the facility even if they are not the ones to do the daily care. This would force producers to vaccinate if no one could take their product without it even if the feedlot wasn't the direct buyer.

Ln 49 I'd also say the paper addresses some ways in which these problems can be addressed.

Ln 51 #3 would be better to be added to #1 as it's not a heading like the other listed problems but examples are provided in individual sections

Ln 53 Move this sentence so the paragraph only discusses the purpose of the paper.

Ln 63 "he" should be "the"

Ln 68 Why do you think feedlots are moving back to concrete floors? It does seem like every 30+ years the industry is cyclic, but consumer demands now would seem to drive towards dirt, natural lots.

Ln 73 need an "improvement" section for lameness as you have for each of the other sections

Ln 98 first sentence punctuation

Ln 149 What decade was this?

Ln 166 Keep this paragraph with the previous as it's just an example of the same heading and not a new category. New heading not needed.

A concluding paragraph is needed to tie the body all together

Author Response

Reviewer 2

Line 19 – Changed wording

Abstract added line on fixing problems

Line 28 – Added transition sentence

Line 41 – Discussed issue of feedlot entry requirements

Line 49 – Added sentence on ways to remedy problems

Line 51 – Moved sentence

Line 63 – Corrected he to the

Line 68 – Added explanation

Line 98 – Corrected punctuation

Line 149 – Added sentence explaining that the lameness study was published in 2012

Line 166 – Removed heading

Reviewer 3 Report

This is a review article which makes the point that although great strides have been made in farm animal welfare, specific problems continue to exist. These problems are summarized as lack of financial responsibility for losses, failure to measure welfare problems, and repeating old mistakes. The concept of “bad becoming normal” is also discussed where lack of financial responsibility, combined with lack of routine use of hard measurement tools by managers can result in non-recognition and acceptance of compromised animal welfare. Methods to improve welfare through management are presented.

Aside from a few grammatical and wording issues, the manuscript is well written. I have no major concerns or suggestions to add.

Specific comments:

Line 16: “…housing fattening cattle for long periods…”

Line 68: “At that time…”

Line 149: “Lameness in dairy cows…that a quarter of all cows are classified as lame (28) and 33% are at risk of becoming lame (29).”

Line 195: “…Holstein are half of the fed cattle market.”

Author Response

Reviewer 3

Corrected the four grammatical mistakes per Reviewer 3’s suggestions

Temple Grandin
